# Second Rényi entropy and annulus partition function for one-dimensional quantum critical systems with boundaries

Benoit Estienne, Yacine Ikhlef, Andrei Rotaru

Sorbonne Université, CNRS, Laboratoire de Physique Théorique et Hautes Énergies, LPTHE, F-75005 Paris, France

May 18, 2022

## Abstract

We consider the entanglement entropy in critical one-dimensional quantum systems with open boundary conditions. We show that the second Rényi entropy of an interval *away from the boundary* can be computed exactly, provided the same conformal boundary condition is applied on both sides. The result involves the annulus partition function. We compare our exact result with numerical computations for the critical quantum Ising chain with open boundary conditions. We find excellent agreement, and we analyse in detail the finite-size corrections, which are known to be much larger than for a periodic system.

## 1 Introduction

The study of quantum entanglement has become a central research field in theoretical high-energy and condensed-matter physics. While the initial motivations stemmed from black hole physics and the holographic principle [1–4], entanglement now finds important applications in low-energy physics, such as the development of tensor network algorithms to simulate strongly-correlated quantum systems [5–7]. More generally, entanglement is a very powerful probe of quantum many-body physics. It can detect phase transitions, extract the central charge and critical exponents of critical points in one-dimensional systems [8–11]. In two dimensions, the entanglement entropy can detect and count critical Dirac fermions [12–15] as well as intrinsic topological order, and extract the various anyonic quantum dimensions [16, 17]. It can also uncover and identify gapless interface modes in both two [18–22] and higher dimensions [23, 24].

Given a quantum system in a pure state $|\Psi\rangle$, the entanglement between two complementary subregions $A$ and $B$ is encoded in the reduced density matrix $\rho_A = \mathrm{Tr}_B |\Psi\rangle\langle\Psi|$. The amount of entanglement can be quantified by entanglement entropies, such as the von Neumann entropy

$$S(A) = -\mathrm{Tr}_A \left( \rho_A \log \rho_A \right) , \tag{1.1}$$

or the *Rényi entropies* [25–29]

$$S_n(A) = \frac{1}{1-n} \log \mathrm{Tr}_A \left( \rho_A^n \right) , \tag{1.2}$$

where $n$ can be any complex number. In particular, the full spectrum of $\rho_A$, called the *entanglement spectrum*, can be recovered from the knowledge of all Rényi entropies [30]. While most of the focus on entanglement entropies has been theoretical, in the past few years there have been many experimental proposals as well as actual experiments to measure them [31–36].

Computing entanglement entropies for strongly-correlated quantum systems is typically a difficult problem. However, if the system is one-dimensional and critical, the full power of two-dimensional Conformal Field Theory (CFT) can be brought to bear. Perhaps the most famous result in this context is the universal asymptotic behaviour [3, 8, 9, 37–39]

$$S_n(\ell) \sim \frac{c}{6} \frac{n+1}{n} \log \ell, \qquad (\ell \to \infty), \tag{1.3}$$

for the entanglement entropy of an interval of length $\ell$ in an infinite system (in the above, $c$ is the central charge of the critical system). CFT computations of entanglement entropies rest on two important insights. First, for integer values of $n$, and if $A$ is the union of some disjoint intervals, the quantity $\text{Tr}_A(\rho_A^n)$ can be expressed as a partition function on an $n$-sheeted Riemann surface with conical singularities corresponding to the endpoints of the intervals [3, 9]. Such partition functions are difficult to evaluate in general, although important results have been obtained for free theories and other particular cases [10, 11, 40–48]. This is where the second insight becomes crucial. Borrowing a trick from the high-energy physics toolbox of the 1980's, one trades the replication of the *spacetime* of the theory to a replication of the *field space* of the CFT [49–51]. Such a construction, known in the literature as the *cyclic orbifold CFT* [50], consists of $n$ copies of the original CFT (referred to as *the mother CFT*), which are modded by the discrete group $\mathbb{Z}_n$ of cyclic permutations. The conical singularities of the mother CFT are accounted for by insertions of *twist operators* [49] in orbifold correlators. Thus, the evaluation of $\text{Tr}_A(\rho_A^n)$ becomes a matter of computing correlators of twist operators. This orbifold approach is very general and flexible, as twist operators can be easily adapted to encode modified initial conditions around the branch points [52], which is relevant for instance in the case of non-unitary systems [53, 52] or for symmetry-resolved entanglement entropy [54–58].

In this article, we consider the entanglement entropy in an open system, when the subregion $A$ is a single interval *away from the boundary*. In the scaling limit, such an open critical system is described by a Boundary Conformal Field Theory (BCFT), with a well-established [59–61] correspondence between the chiral Virasoro representations and the *conformal boundary conditions* (BC) allowed by the theory. The case of an interval touching the boundary has been extensively studied (see [62] for a review) using either conformal field theory methods [9, 39, 63–65] or exact free fermion methods [66–68], including symmetry-resolved entropies [69]. It has also been checked numerically using density-matrix renormalization group techniques [70–73]. When the subregion $A$ is an interval at the end of a semi-infinite line, or at the end of a finite system with the same boundary condition on both sides, the computation of the Rényi entanglement entropy boils down to the evaluation of a twist one-point function on the upper half-plane. Such a correlation function is simply fixed by conformal invariance, and as a consequence the entanglement entropy exhibits again a simple space dependence, similarly to (1.3). For instance, in the case of an interval of length $\ell$ at the end of a semi-infinite line one finds [1] [9]

$$S_n(\ell) \sim \frac{c}{12} \frac{n+1}{n} \log 2\ell + \log g_\alpha, \qquad (\ell \to \infty), \tag{1.4}$$

where $g_\alpha$ is the *universal boundary entropy* [74] associated to the boundary condition $\alpha$.

In contrast, there are very few results for an interval *away from the boundary*, mainly because the CFT computation is much more involved. Indeed, after a proper conformal mapping, one has

---

[1] up to an additive non-universal constant coming from the normalization of the lattice twist operator

to compute a two-twist correlation function in the upper half-plane, which is no longer a simple power law fixed by conformal symmetry. As a consequence, the corresponding entanglement entropy is generally not known, despite some partial recent results for free scalar fields [75,76], or in the large central charge limit [77]. In this article, we report an exact computation of the second Rényi entropy $S_2$ of a single interval in the ground state of a 1D critical system with open boundaries, assuming the same boundary conditions on both sides: see Fig. 1.

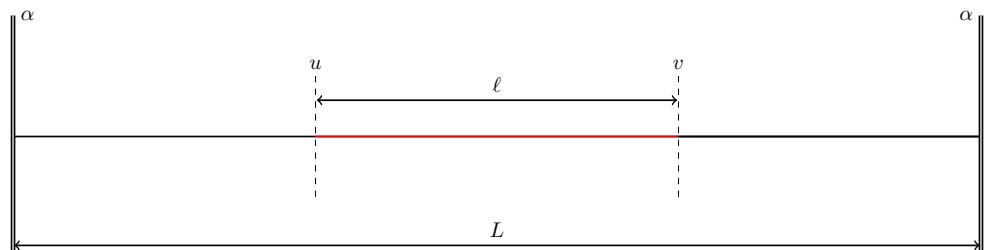

Figure 1: A generic bipartition of a 1D system with boundary condition $\alpha$ on both ends.

As mentioned above, the calculation rests on the evaluation of a two-point function of twist operators on the infinite strip. With the restriction that the same conformal boundary condition $\alpha$ is chosen on both sides of the system, this boils down to the computation of the two-twist correlation function on the unit disk $\mathbb{D}$ (with no boundary operator inserted). The main result of this paper is the computation of this two-twist correlation function in terms of the annulus partition function of the mother CFT:

$$\langle \sigma(0)\sigma(x,\bar{x})\rangle_{\mathbb{D}}^{(\alpha,\alpha)} = g_\alpha^{-2}\, 2^{-\frac{c}{3}} \left[|x|^2(1-|x|^2)\right]^{-\frac{c}{24}} Z_{\alpha|\alpha}(\tau)\,, \tag{1.5}$$

where $\sigma$ denotes the twist operator in the $\mathbb{Z}_2$ orbifold CFT, $c$ is the central charge and $Z_{\alpha|\alpha}(\tau)$ is the partition function on the annulus of unit circumference, width $\operatorname{Im}\tau/2$, and boundary condition $\alpha$ on both edges. The parameter $\tau$ (which is pure imaginary) is related to $x$ via:

$$\left[\frac{\theta_2(\tau)}{\theta_3(\tau)}\right]^2 = |x|\,, \quad \text{or equivalently} \quad \tau(x,\bar{x}) = i\,\frac{{}_2\mathrm{F}_1\left(\frac{1}{2},\frac{1}{2},1;1-|x|^2\right)}{{}_2\mathrm{F}_1\left(\frac{1}{2},\frac{1}{2},1;|x|^2\right)}\,, \tag{1.6}$$

where the $\theta_j(\tau)$'s are the Jacobi elliptic functions (see appendix A.1). Moreover, the universal boundary entropy $g_\alpha$, appearing in (1.4–1.5), can be simply defined in terms of BCFT states – see (2.26). These results are completely general, and apply to any mother CFT with a known annulus partition function. This includes, of course, CFTs built from minimal models and Wess-Zumino-Witten models [78,79], free and compactified bosonic CFTs [80] to name a few. This result is reminiscent of the well-known relation between the twist four-point function on the sphere and the torus partition function $Z(\tau,\bar{\tau})$ [49,81,82]

$$\langle \sigma(0)\sigma(\eta,\bar{\eta})\sigma(1)\sigma(\infty)\rangle_{\mathbb{C}} = 4^{-\frac{c}{3}} |\eta(1-\eta)|^{-\frac{c}{12}} Z(\tau,\bar{\tau})\,, \qquad \eta = \left[\frac{\theta_2(\tau)}{\theta_3(\tau)}\right]^4\,. \tag{1.7}$$

The final result for the $S_2$ entropy of an interval $A = [u,v]$ in a system of length $L$ is, up to an additive non-universal constant coming from the normalization of the lattice twist operator :

$$\boxed{\mathrm{S}_2^\alpha([u,v]) = \frac{c}{24}\log\left[s_L(2u)s_L(2v)s_L^2(v+u)s_L^2(v-u)\right] + 2\log g_\alpha - \log Z_{\alpha|\alpha}(\tau)\,,} \tag{1.8}$$

where we have introduced the shorthand notation

$$s_L(w) = \frac{2L}{\pi}\sin\frac{\pi w}{2L}\,. \tag{1.9}$$

The parameter $\tau$ is related to the position of the interval $[u, v]$ (where $0 < u < v < L$) via

$$\frac{\sin \frac{\pi(v-u)}{2L}}{\sin \frac{\pi(v+u)}{2L}} = \left[\frac{\theta_2(\tau)}{\theta_3(\tau)}\right]^2 . \tag{1.10}$$

We give here the organization of the paper. Section 2 provides a detailed derivation of the main result (1.8) and a non-trivial check that our calculation does recover the result of [63,9,67] for the $S_2$ Rényi entropy of an interval $A$ touching the boundary. We also recover the known results for the Dirac fermion [66, 83, 84] and more generally the compact scalar field of [75]. Lastly, we extend our results to exact expressions for the mutual information and the entropy distance in specific situations. In Section 3, we present some numerical checks of (1.8) for the particular case of the Ising spin chain, using an efficient numerical method known as *Peschel's trick*, which allows the numerical determination of the entanglement spectrum for fermionic chains of system sizes up to $N \sim 10^3$ sites. We also carry a careful analysis of the finite-size effects. In Section 4, we conclude with a recapitulation of our results and comment on future directions for exploration. The Appendices A, B and C contain respectively our notations and conventions for elliptic functions, an alternate derivation of the main result based on boundary CFT techniques applied to the $\mathbb{Z}_2$ orbifold, and the computation of the bosonic annulus partition function.

## 2 Exact calculation of the second Rényi entropy

We consider a one-dimensional quantum critical system of finite length $L$, with open boundary conditions, and at zero temperature. We are interested in the second Rényi entropy of an interval $[u, v]$. The critical point is assumed to be described by a CFT. For a large enough system, the boundary flows to a renormalisation-group fixed point. We will therefore assume that the boundary condition is scale invariant. For a given bulk universality class, there is a set of possible such conformal boundary conditions $\{B_\alpha\}$ [59], [61], [60], [85]. We restrict to the case where the same boundary conditions are applied at the two ends of the system, and we assume that there is a non-degenerate ground state $|\psi_0\rangle$.

Evaluating the second Rényi entropy $S_2^\alpha([u, v])$ boils down to the computation of the following correlator in the $\mathbb{Z}_2$ orbifold of the original CFT [39], [9], [77], [71]:

$$\langle \sigma(u, \bar{u})\sigma(v, \bar{v})\rangle_{\mathbb{S}_L}^{(\alpha,\alpha)} = \exp[-S_2^\alpha([u, v])] , \tag{2.1}$$

where $\sigma$ denotes the twist operator[2]. This correlator is evaluated on the infinite strip (with imaginary time running along the imaginary axis) $\mathbb{S}_L = \{w \in \mathbb{C}, 0 < \text{Re}(w) < L\}$ of width $L$ with boundary condition $(\alpha, \alpha)$ on both sides of the strip. Alternatively this two-point function is equal to the following ratio of partition functions [3,9]:

$$\langle \sigma(u, \bar{u})\sigma(v, \bar{v})\rangle_{\mathbb{S}_L}^{(\alpha,\alpha)} = Z_2(u, v)/Z_1^2 , \tag{2.2}$$

where $Z_1$ stands for the strip partition function, and $Z_2(u, v)$ stands for the partition function on a two-sheeted covering of the infinite strip with branch points at $u$ and $v$, being understood that all edges have the same conformal boundary condition $\alpha$. The main result of this paper rests on

---

[2]Here, even though $u$ and $v$ are real, and hence $u = \bar{u}$ and $v = \bar{v}$, we use the standard notations $\sigma(u, \bar{u})$ and $\sigma(v, \bar{v})$ for bulk operators, which emphasizes the fact that the correlation function is not a holomorphic function of $u$ and $v$.

the fact that this Riemann surface (once compactified) is conformally equivalent to an annulus, as was observed in [77]. It is therefore not surprising that the two-twist correlation function is equal, up to some universal prefactors, to the annulus partition function. We present two different ways to derive this result. The first method, which we now detail, is more geometric in nature : we unfold the two-sheeted Riemann surface into an annulus via an explicit conformal mapping. The second method, which is more algebraic, is based on Cardy's mirror trick [59,85] applied to the $\mathbb{Z}_2$ orbifold. This second approach, which employs a larger set of BCFT and orbifold concepts, has been relegated to Appendix B to avoid congesting the logical flow of the article.

## 2.1 Conformal equivalence to the annulus

In order to construct an explicit conformal map between the two-sheeted strip and the annulus, it is convenient to first map the strip to the unit disk via

$$w \mapsto z = \frac{s_L(w - u)}{s_L(w + u)}, \qquad s_L(w) = \frac{2L}{\pi} \sin \frac{\pi w}{2L}. \tag{2.3}$$

The above conformal map also sends the two-sheeted strip (with branch points at $w = u$ and $w = v$) to the two-sheeted unit disk $\mathbb{D}_{2,x}$ with branch points at $z = 0$ and $z = x$, with

$$x = \frac{s_L(v - u)}{s_L(v + u)} = \frac{\sin \frac{\pi}{2L}(v - u)}{\sin \frac{\pi}{2L}(v + u)}. \tag{2.4}$$

Note that $x$ is real, and $0 < x < 1$. Let us now describe the conformal mapping sending $\mathbb{D}_{2,x}$ to an annulus. First, for any complex number $\tau$ with $\operatorname{Im} \tau > 0$, the function

$$t \mapsto z = g(t) = \left( \frac{\theta_4(t|\tau)}{\theta_1(t|\tau)} \right)^2 \tag{2.5}$$

is a biholomorphic map from the torus of modular parameter $\tau$ to the double-sheeted cover of the Riemann sphere with four branch points at positions

$$g(0) = \infty, \quad g\left( \frac{1+\tau}{2} \right) = x, \quad g\left( \frac{\tau}{2} \right) = 0, \quad g\left( \frac{1}{2} \right) = \frac{1}{x}, \tag{2.6}$$

with

$$x = \left( \frac{\theta_2(\tau)}{\theta_3(\tau)} \right)^2, \qquad \tau = \mathrm{i} \frac{{}_2\mathrm{F}_1\left( \frac{1}{2}, \frac{1}{2}, 1; 1 - x^2 \right)}{{}_2\mathrm{F}_1\left( \frac{1}{2}, \frac{1}{2}, 1; x^2 \right)}, \tag{2.7}$$

and where the $\theta_j(t, q)$'s are Jacobi theta functions (see Appendix A.1 for definitions and conventions). Using the properties (A.3) of these functions, we readily see that the function $g$ satisfies the identity:

$$g(t + \tau/2) = g(t)^{-1}, \tag{2.8}$$

for any $t$ on the torus. In the present situation, since $0 < x < 1$, the modular parameter $\tau$ is pure imaginary, with $\operatorname{Im} \tau > 0$. Then, from the above relation we get

$$g\left( \frac{\tau}{2} + \bar{t} \right) = \overline{g(t)^{-1}}. \tag{2.9}$$

Now notice that identifying $t$ and $\tau/2 + \bar{t}$ amounts to folding the torus into an annulus of unit width, and height $\operatorname{Im} \tau/2$

$$\mathbb{A}_\tau = \left\{ t \in \mathbb{C}/\mathbb{Z}, \qquad \frac{\operatorname{Im} \tau}{4} \leq \operatorname{Im} t \leq \frac{3 \operatorname{Im} \tau}{4} \right\}, \tag{2.10}$$

while identifying $z$ and $1/\bar{z}$ on the two-sheeted Riemann sphere yields the two-sheeted unit disk $\mathbb{D}_{2,x}$. In essence, these foldings are the reverse of Cardy's mirror trick [59]. The relation (2.9) ensures that the map $g$ descends to the quotient, yielding a biholomorphic map from the annulus $\mathbb{A}_\tau$ to the two-sheeted unit disk $\mathbb{D}_{2,x}$, as shown in Figure 2.

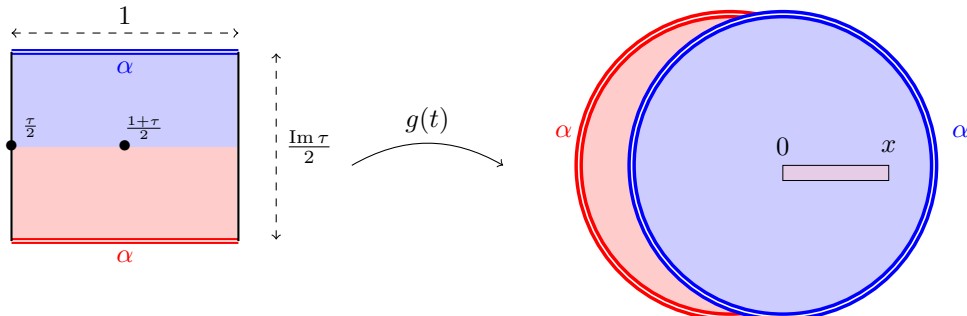

Figure 2: The annulus $\mathbb{A}_\tau$ – *fundamental domain pictured here* – is mapped through $g$ to the two-sheeted disk $\mathbb{D}_{2,x}$. The black edges are identified.

## 2.2   Rényi entropy of an interval in the bulk

Recall that the twist $\sigma$ is a primary operator of conformal dimensions $h_\sigma = \bar{h}_\sigma = c/16$ in the $\mathbb{Z}_2$ orbifold CFT. Using conformal covariance under the map (2.3), we can relate the twist correlation functions on the strip and on the unit disk:

$$\langle \sigma(u,\bar{u})\sigma(v,\bar{v})\rangle_{\mathbb{S}_L}^{(\alpha,\alpha)} = (s_L(u+v))^{-c/4}\,\langle\sigma(0,0)\sigma(x,\bar{x})\rangle_{\mathbb{D}}^{(\alpha,\alpha)}\,, \tag{2.11}$$

where

$$x = \bar{x} = \frac{s_L(v-u)}{s_L(u+v)} \geq 0\,. \tag{2.12}$$

The strategy (adapted from [49]) to compute $\langle\sigma(0,0)\sigma(x,\bar{x})\rangle_{\mathbb{D}}^{(\alpha,\alpha)}$ in terms of an annulus partition function is the following. We insert the stress-energy tensor $T_{\mathrm{orb}}$ into the twist correlation function on the unit disk, and study the behaviour of the function $\langle T_{\mathrm{orb}}(z)\sigma(0,0)\sigma(x,\bar{x})\rangle_{\mathbb{D}}^{(\alpha,\alpha)}$ as $z \to x$. Since $\sigma$ is a primary operator, we have the OPE

$$T_{\mathrm{orb}}(z)\sigma(x,\bar{x}) = \frac{h_\sigma \sigma(x,\bar{x})}{(z-x)^2} + \frac{\partial_x \sigma(x,\bar{x})}{z-x} + \text{regular terms}, \tag{2.13}$$

and thus

$$\partial_x \log\langle\sigma(0,0)\sigma(x,\bar{x})\rangle_{\mathbb{D}}^{(\alpha,\alpha)} = \frac{1}{2\pi i}\oint_{\mathcal{C}_x} \frac{\langle T_{\mathrm{orb}}(z)\sigma(0,0)\sigma(x,\bar{x})\rangle_{\mathbb{D}}^{(\alpha,\alpha)}}{\langle\sigma(0,0)\sigma(x,\bar{x})\rangle_{\mathbb{D}}^{(\alpha,\alpha)}}\,dz\,, \tag{2.14}$$

where the integration contour $\mathcal{C}_x$ encloses the point $x$ and goes anti-clockwise. However, in (2.13) and (2.14) the parameter $x$ stands for a complex variable (independent of $\bar{x}$). Setting $x = \bar{x}$ thus yields

$$\partial_x \left(\log\langle\sigma(0,0)\sigma(x,\bar{x})\rangle_{\mathbb{D}}^{(\alpha,\alpha)}\Big|_{x=\bar{x}}\right) = 2 \times \frac{1}{2\pi i}\oint_{\mathcal{C}_x} \frac{\langle T_{\mathrm{orb}}(z)\sigma(0,0)\sigma(x,\bar{x})\rangle_{\mathbb{D}}^{(\alpha,\alpha)}}{\langle\sigma(0,0)\sigma(x,\bar{x})\rangle_{\mathbb{D}}^{(\alpha,\alpha)}}\,dz\,. \tag{2.15}$$

We will drop the $|_{x=\bar{x}}$, but from now on $x$ is assumed – without loss of generality – to be real positive, with $0 < x < 1$.

In terms of the mother theory, $\langle T_{\mathrm{orb}}(z)\sigma(0,0)\sigma(x,\bar{x})\rangle_{\mathbb{D}}^{(\alpha,\alpha)}$ is the one-point function of the stress-energy tensor on the two-sheeted surface $\mathbb{D}_{2,x}$. Since $T_{\mathrm{orb}}(z) = T(z) \otimes \mathbb{I} + \mathbb{I} \otimes T(z)$, we can write

$$\frac{\langle T_{\mathrm{orb}}(z)\sigma(0,0)\sigma(x,\bar{x})\rangle_{\mathbb{D}}^{(\alpha,\alpha)}}{\langle \sigma(0,0)\sigma(x,\bar{x})\rangle_{\mathbb{D}}^{(\alpha,\alpha)}} = 2\langle T(z)\rangle_{\mathbb{D}_{2,x}}^{\alpha}\,, \tag{2.16}$$

where the last equality comes from the symmetry under the exchange of the two copies of the unit disk. The last step is to compute $\langle T(z)\rangle_{\mathbb{D}_{2,x}}^{\alpha}$ by exploiting the conformal equivalence between the two-sheeted cover of the disk $\mathbb{D}_{2,x}$ and the annulus $\mathbb{A}_{\tau}$ via the map $z = g(t)$ described in (2.5):

$$\langle T(z)\rangle_{\mathbb{D}_{2,x}}^{\alpha} = \left(\frac{dt}{dz}\right)^2 \langle T(t)\rangle_{\mathbb{A}_{\tau}}^{\alpha} + \frac{c}{12}\{t,z\}\,, \tag{2.17}$$

where $\{t,z\}$ denotes the Schwarzian derivative of the map $g$. First, the one-point function of $T(z)$ on the annulus is

$$\langle T(t)\rangle_{\mathbb{A}_{\tau}}^{\alpha} = 2i\pi\partial_{\tau} \log Z_{\alpha|\alpha}(\tau)\,, \tag{2.18}$$

where $Z_{\alpha|\alpha}(\tau)$ denotes the partition function on the annulus $\mathbb{A}_{\tau}$ (with boundary condition $\alpha$ on both edges). Let $|\alpha\rangle$ be the boundary state associated to the boundary condition $\alpha$. Since $\mathbb{A}_{\tau}$ has unit width, and height $\beta/2 = -i\tau/2$, we have

$$Z_{\alpha|\alpha}(\tau) = \langle\alpha|e^{i\pi\tau(L_0+\bar{L}_0-c/12)}|\alpha\rangle\,. \tag{2.19}$$

We can exploit the differential equation (A.17) obeyed by the map $z = g(t)$, namely

$$\left(\frac{dt}{dz}\right)^2 = -\frac{1}{4\pi^2\theta_3^4(\tau)z(z-x)(1-xz)}\,, \tag{2.20}$$

to derive

$$\langle T(z)\rangle_{\mathbb{D}_{2,x}}^{\alpha} = \frac{x(1-x^2)}{4z(z-x)(1-zx)}\partial_x \log Z_{\alpha|\alpha}(\tau) + \frac{c}{12}\{t,z\}\,, \tag{2.21}$$

where we have also used the relation (A.11). The Schwarzian derivative can be easily evaluated using (2.20), yielding

$$\{t,z\} = \frac{3x^2(1+z^4) - 4\left(x+x^3\right)\left(z+z^3\right) + 2\left(2x^4+x^2+2\right)z^2}{8z^2(z-x)^2(1-xz)^2}\,, \tag{2.22}$$

and in particular the residue at $z \to x$ is

$$\frac{1}{2\pi i}\oint_{\mathcal{C}_x}\{t,z\}\,dz = -\frac{1-2x^2}{4x\left(1-x^2\right)} = -\frac{1}{8}\partial_x \log x^2(1-x^2)\,. \tag{2.23}$$

Finally plugging the above in (2.15) we get

$$\partial_x \log\langle\sigma(0,0)\sigma(x,\bar{x})\rangle_{\mathbb{D}}^{(\alpha,\alpha)} = \partial_x \log Z_{\alpha|\alpha}(\tau) - \frac{c}{24}\partial_x \log x^2(1-x^2)\,. \tag{2.24}$$

Upon integration, we obtain

$$\langle\sigma(0,0)\sigma(x,\bar{x})\rangle_{\mathbb{D}}^{(\alpha,\alpha)} = \mathrm{const} \times \left[x^2(1-x^2)\right]^{-\frac{c}{24}} Z_{\alpha|\alpha}(\tau)\,. \tag{2.25}$$

In order to fix the multiplicative constant in the above relation, we consider the leading behaviour as $x$ tends to zero. In this limit, we have $\mathrm{Im}\,\tau \to +\infty$ and $q \to 0$, with the relation $q = e^{2i\pi\tau} \sim (x/4)^4$. Thus

$$Z_{\alpha|\alpha}(\tau) = \langle\alpha|e^{i\pi\tau(L_0+\bar{L}_0-c/12)}|\alpha\rangle \underset{\mathrm{Im}\,\tau\to\infty}{\sim} q^{-c/24}\,g_{\alpha}^2\,, \qquad g_{\alpha} = |\langle\alpha|0\rangle|\,, \tag{2.26}$$

where $|0\rangle$ is the normalized ground state wavefunction of the Hamiltonian with periodic boundary conditions. The twist operator $\sigma$ is normalized so that

$$\langle\sigma(0,0)\sigma(x,\bar{x})\rangle_{\mathbb{D}}^{(\alpha,\alpha)} \underset{x\to 0}{\sim} x^{-c/4}\,, \tag{2.27}$$

and hence the fully explicit relation (2.25) is

$$\boxed{\langle\sigma(0,0)\sigma(x,\bar{x})\rangle_{\mathbb{D}}^{(\alpha,\alpha)} = g_\alpha^{-2}\, 2^{-\frac{c}{3}} \left[x^2(1-x^2)\right]^{-\frac{c}{24}} Z_{\alpha|\alpha}(\tau)\,.} \tag{2.28}$$

Note that in the above equation we have assumed $x = \bar{x}$, with $0 < x < 1$. For a generic complex $x$ on the unit disk, the result still holds up to replacing $x$ by $|x|$ in the *r.h.s.* as well as in (2.7). Back to the original problem on the strip, we obtain

$$\langle\sigma(u,\bar{u})\sigma(v,\bar{v})\rangle_{\mathbb{S}_L}^{(\alpha,\alpha)} = g_\alpha^{-2} 2^{-\frac{c}{3}} \left[s_L(v+u)^2 s_L(v-u)^2 s_L(2u) s_L(2v)\right]^{-\frac{c}{24}} Z_{\alpha|\alpha}(\tau)\,, \tag{2.29}$$

and we get the announced result (1.8) for the second Rényi entropy.

## 2.3   Rényi Entropy of an interval touching the boundary

As a check for the formula (2.29), we want to recover the expression for the Rényi entropy $S_2$ of an interval $A = [0, \ell]$ touching the boundary of the chain [9, 70, 86, 71, 63]:

$$S_2^\alpha([0,\ell]) = \frac{c}{8} \log\left[\frac{2L}{\pi} \sin\left(\frac{\pi\ell}{L}\right)\right] + \log g_\alpha\,. \tag{2.30}$$

Let us consider the two-point function $\langle\sigma(u,\bar{u})\sigma(v,\bar{v})\rangle_{\mathbb{S}_L}^{(\alpha,\alpha)}$ in the limit $u \to 0$. On the left-hand side of (2.29), we can use the bulk-boundary OPE:

$$\sigma(u,\bar{u}) \underset{u\to 0}{\sim} A_\sigma^\alpha\, (u+\bar{u})^{-c/8}\,\mathbb{I}\,, \tag{2.31}$$

where $A_\sigma^\alpha = (g_\alpha)^{-1}$ is the OPE coefficient for the bulk operator $\sigma$ approaching a boundary with boundary condition $\alpha$, and giving rise to the boundary identity operator $\mathbb{I}$. Hence, for $u$ real:

$$\langle\sigma(u,\bar{u})\sigma(\ell)\rangle_{\mathbb{S}_L}^{(\alpha,\alpha)} \underset{u\to 0}{\sim} (g_\alpha)^{-1} (2u)^{-c/8} \langle\sigma(\ell)\rangle_{\mathbb{S}_L}^{(\alpha,\alpha)}\,. \tag{2.32}$$

On the right-hand side of (2.29), the limit $u \to 0$ corresponds to $x \to 1$ and $\tau \to 0$, with

$$\widetilde{q} = e^{-2i\pi/\tau} \sim \left(\frac{1-x^2}{16}\right)^2 \to 0\,. \tag{2.33}$$

In a rational CFT, the annulus partition function decomposes on the characters of primary representations $V_k$ as

$$Z_{\alpha|\alpha}(\tau) = \sum_k n_{\alpha\alpha}^k\, \chi_k(-1/\tau)\,, \qquad \chi_k(\tau) = \mathrm{Tr}_{V_k}\left(q^{L_0-c/24}\right)\,. \tag{2.34}$$

In the limit $\widetilde{q} \to 0$, we get $Z_{\alpha|\alpha}(\tau) \sim n_{\alpha\alpha}^0 \widetilde{q}^{-c/24}$, where $k = 0$ stands for the identity operator, and $n_{\alpha\alpha}^0 = 1$, since we assumed a non-degenerate ground state $|\psi_0\rangle$. Thus

$$Z_{\alpha|\alpha}(\tau) \sim \left(\frac{1-x^2}{16}\right)^{-c/12} \sim 2^{\frac{c}{4}} \left(\frac{s_L^2(v)}{u\, s_L(2v)}\right)^{\frac{c}{12}}\,, \tag{2.35}$$

where we have used (2.4) to obtain the second relation. After some simple algebra, one gets for the right-hand side of (2.29):

$$\langle \sigma(u)\sigma(v) \rangle^{(\alpha,\alpha)}_{\mathbb{S}_L} \underset{u \to 0}{\sim} g_\alpha^{-2} \left( \frac{1}{\frac{2L}{\pi} \sin \frac{\pi \ell}{L}} \right)^{c/8} (2u)^{-c/8} . \tag{2.36}$$

Hence, comparing (2.32) and (2.36), we recover the well known one-point function

$$\langle \sigma(\ell) \rangle^{(\alpha,\alpha)}_{\mathbb{S}_L} = g_\alpha^{-1} \left( \frac{1}{\frac{2L}{\pi} \sin \frac{\pi \ell}{L}} \right)^{c/8} , \tag{2.37}$$

which indeed yields (2.30).

## 2.4 Compact boson

We can further apply our main formula (1.8) by recovering the known results for the Dirac fermion [66,83,84] and more generally the compact boson [75]. We consider a compact scalar field $\phi \equiv \phi + 2\pi R$ with action

$$S[\phi] = \frac{1}{8\pi} \int d^2 r \, \partial_\mu \phi \, \partial^\mu \phi \,, \tag{2.38}$$

and Dirichlet boundary conditions. The relevant annulus partition function is (see Appendix C)

$$Z(\tau) = \frac{\theta_3(-R^2/\tau)}{\eta(-1/\tau)} . \tag{2.39}$$

Before plugging this partition function into our main formula (1.8), let us write it as

$$Z(\tau) = \frac{\theta_3(-1/\tau)}{\eta(-1/\tau)} \times \frac{\theta_3(-R^2/\tau)}{\theta_3(-1/\tau)} = \frac{\theta_3(\tau)}{\eta(\tau)} \times \frac{\theta_3(-R^2/\tau)}{\theta_3(-1/\tau)} . \tag{2.40}$$

Now using

$$\frac{\theta_3(\tau)}{\eta(\tau)} = 2^{1/3} \left[ x^2(1-x^2) \right]^{-\frac{1}{12}} = 2^{\frac{1}{3}} \left( \frac{s_L^2(v-u)s_L(2u)s_L(2v)}{s_L^4(v+u)} \right)^{-\frac{1}{12}} . \tag{2.41}$$

we find (up to an additive constant)

$$\mathrm{S}_2^\alpha([u,v]) = \frac{1}{8} \log \frac{s_L(2u)s_L(2v)s_L^2(v-u)}{s_L^2(v+u)} - \log \mathcal{F}_2(\tau) \,, \tag{2.42}$$

where the function $\mathcal{F}_2(\tau)$ is given by

$$\mathcal{F}_2(\tau) = \frac{\theta_3(-R^2/\tau)}{\theta_3(-1/\tau)} = \frac{\sum\limits_{m \in \mathbb{Z}} \exp\left(-i\pi m^2 R^2/\tau\right)}{\sum\limits_{m \in \mathbb{Z}} \exp\left(-i\pi m^2/\tau\right)} . \tag{2.43}$$

This is equivalent to the formulae (13) and (18) of [75] provided $\bar{\mathcal{M}} = i\pi/4\tau$ in (18) [although we note a typo in the first term of (13)], and the result for the Dirac fermion [namely $\mathcal{F}_2(\tau) = 1$ for $R = 1$] follows.

## 2.5   Other entanglement measures

In this section, we present two other entanglement measures related to the second Rényi entropy: mutual information and entropy distance. Here they are defined in the same context as considered above, namely in a critical 1d quantum system of finite size $L$, with open boundaries, and the same conformal boundary condition on both sides.

When considering two disjoint subsystems $A$ and $B$, a standard measure of the information "shared" by $A$ and $B$ is given by the *mutual information $I_{A:B}$*, defined as (see [11] and references therein)

$$I_{A:B} = S_A + S_B - S_{A \cup B}, \tag{2.44}$$

where $S$ stands for a given measure of entanglement for a single subsystem. Using our result (1.8), we can express the mutual information (associated to the second Rényi entropy) of two intervals each touching a different boundary of the system, namely $A = [0, u]$ and $B = [v, L]$. After some straightforward algebra on (1.8) and (2.30), we get

$$I_{[0,u]:[v,L]} = \frac{c}{12} \log \left[ \frac{s(2u)s(2v)}{s(v+u)s(v-u)} \right] + \log Z_{\alpha|\alpha}(\tau). \tag{2.45}$$

Back to the situation of a subsystem $A$ consisting of a single interval $[u, v]$ inside the bulk of the system, we turn to the question of quantifying how much the whole spectrum of the density matrix $\rho_A$ depends on the choice of external parameters (see [87] and references therein) – in the present case, the external parameter is the boundary condition. Here, we shall use the $n$-norm of an operator $\Lambda$, defined as

$$\|\Lambda\|_n = \left\{ \mathrm{Tr} \left[ (\Lambda^\dagger \Lambda)^{n/2} \right] \right\}^{1/n}, \tag{2.46}$$

and the associated Schatten distance[3]

$$D_n(\rho, \rho') = \|\rho - \rho'\|_n. \tag{2.47}$$

To be specific, we denote by $\rho_{A,\alpha}$ the reduced density matrix associated to the ground state of our finite critical systems with boundary conditions $\alpha$ on both sides of the system. Then we consider the Schatten distance $D_n(\rho_{A,\alpha}, \rho_{A,\beta})$, where $\alpha$ and $\beta$ are two distinct conformal BCs. We restrict to the value $n = 2$, and we have

$$D_2(\rho_{A,\alpha}, \rho_{A,\beta}) = \left[ \frac{1}{2} \mathrm{Tr}\, \rho_{A,\alpha}^2 + \frac{1}{2} \mathrm{Tr}\, \rho_{A,\beta}^2 - \mathrm{Tr}(\rho_{A,\alpha}\rho_{A,\beta}) \right]^{1/2}. \tag{2.48}$$

The first two terms in (2.48) are given by (1.8), whereas the third term is obtained by a slight generalization of the previous discussion. Indeed, we can write this term as the two-twist correlation function

$$\mathrm{Tr}(\rho_{A,\alpha}\rho_{A,\beta}) = \langle \sigma(u, \bar{u})\sigma(v, \bar{v}) \rangle_{\mathbb{S}_L}^{(\alpha,\beta)} \tag{2.49}$$

on the infinite strip with BC $\alpha$ (resp. $\beta$) on both sides, for the first (resp. second) copy of the mother CFT in the $\mathbb{Z}_2$ orbifold. Through the same line of argument as in Section 2.2, we obtain

$$\langle \sigma(u, \bar{u})\sigma(v, \bar{v}) \rangle_{\mathbb{S}_L}^{(\alpha,\beta)} = (g_\alpha g_\beta)^{-1} 2^{-\frac{c}{3}} \left[ s_L(v+u)^2 s_L(v-u)^2 s_L(2u) s_L(2v) \right]^{-\frac{c}{24}} Z_{\alpha|\beta}(\tau). \tag{2.50}$$

---

[3]While the most interesting distance is $D_1$, it can be extremely difficult to evaluate directly. One can instead exploit a replica trick developed in [88, 89]: one first computes the distance $D_n$ for all even $n$, followed by an analytic continuation to $n = 1$.

As a result, we get

$$D_2(\rho_{A,\alpha}, \rho_{A,\beta}) = 2^{-\frac{c}{6}} \left[ s_L(v+u)^2 s_L(v-u)^2 s_L(2u) s_L(2v) \right]^{-\frac{c}{48}} K_{\alpha\beta}(\tau), \tag{2.51}$$

where

$$K_{\alpha\beta}(\tau) = \left[ \frac{Z_{\alpha|\alpha}(\tau)}{2g_\alpha^2} + \frac{Z_{\beta|\beta}(\tau)}{2g_\beta^2} - \frac{Z_{\alpha|\beta}(\tau)}{g_\alpha g_\beta} \right]^{1/2}. \tag{2.52}$$

Plugging in the expression of the annulus partition function in terms of Ishibashi states (B.9)

$$Z_{\alpha|\beta}(\tau) = \langle \alpha | e^{i\pi\tau(L_0 + \bar{L}_0 - c/12)} | \beta \rangle = \sum_j (\Psi_j^\alpha)^* \Psi_j^\beta \chi_j(\tau) \tag{2.53}$$

yields

$$K_{\alpha\beta}(\tau) = \left[ \frac{1}{2} \sum_j \left| A_j^\alpha - A_j^\beta \right|^2 \chi_j(\tau) \right]^{1/2}. \tag{2.54}$$

Interestingly the term $j = 0$ cancels out, as follows from $A_0^\alpha = A_0^\beta = 1$. This means that the vacuum sector does not contribute to the Schatten distance. Furthermore this last expression is rather suggestive : it is the distance associated to the following $L^2$ norm (weighted by the positive coefficients $\chi_j(\tau)/2$) on the $A_j$ space

$$\|A\| = \left[ \frac{1}{2} \sum_j |A_j|^2 \chi_j(\tau) \right]^{1/2}. \tag{2.55}$$

We shall now consider some limiting cases of the Schatten distance (2.51).

### 2.5.1  Small interval in the bulk

The limit of a very small interval in the bulk is recovered for $u \to v$, which corresponds to $q = e^{2i\pi\tau} \to 0$. In this regime we have

$$\chi_j(\tau) \sim q^{h_j - c/24} \tag{2.56}$$

As mentioned above the term $j = 0$ does not contribute, so the $L^2$ norm (2.55) is dominated by the term $j_0$ corresponding to the most relevant state such that

$$A_{j_0}^\alpha \neq A_{j_0}^\beta. \tag{2.57}$$

Then

$$K_{\alpha\beta}(\tau) \sim \frac{1}{\sqrt{2}} \left| A_{j_0}^\alpha - A_{j_0}^\beta \right| q^{h_{j_0}/2 - c/48}. \tag{2.58}$$

and in the limit of a small interval in the bulk ($\ell \to 0$) the Schatten distance behaves, up to a constant prefactor, as

$$D_2(\rho_{A,\alpha}, \rho_{A,\beta}) \underset{\ell \to 0}{\sim} \ell^{2h_{j_0} - c/8} \frac{\left| A_{j_0}^\alpha - A_{j_0}^\beta \right|}{s_L(2v)^{2h_{j_0}}}. \tag{2.59}$$

### 2.5.2  Interval touching the boundary

In the limit $u \to 0$ ($v$ fixed), the parameter $q$ goes to 1 so it is more convenient to work with $\tilde{q} = e^{-2i\pi/\tau}$. Thus we use expression (2.52) together with

$$Z_{\alpha|\beta}(\tau) = \sum_k n_{\alpha\beta}^k \chi_k(-1/\tau), \qquad \chi_k(-1/\tau) = \text{Tr}_{V_k}\left(\tilde{q}^{L_0 - c/24}\right) \underset{\tilde{q}\to 0}{\sim} \tilde{q}^{h_k - c/24}, \tag{2.60}$$

For the vacuum sector to propagate, the left and right conformal boundary conditions must be the same [79] :

$$n_{\alpha\beta}^0 = \delta_{\alpha\beta} \tag{2.61}$$

This implies that the identity character $\chi_0$ does not appear in the expansion of $Z_{\alpha|\beta}(\tau)$ for $\alpha \neq \beta$. Thus, the leading order behaviour of the annulus partition function is:

$$Z_{\alpha|\beta}(\tau) \sim \tilde{q}^{-c/24 + h_{k_0}} \tag{2.62}$$

where $k_0$ corresponds to the most relevant state that can propagate with boundary conditions $\alpha$ on one side and $\beta$ on the other. Equivalently, $h_{k_0}$ is the lowest allowed conformal dimension in the spectrum of boundary changing operators between $\alpha$ and $\beta$. This implies that for an interval strictly touching the boundary the states $\rho_{A,\alpha}$ and $\rho_{A,\beta}$ simply become orthogonal

$$D_2(\rho_{A,\alpha}, \rho_{A,\beta}) \underset{u\to 0}{\to} \sqrt{\|\rho_{A,\alpha}\|^2 + \|\rho_{A,\beta}\|^2}. \tag{2.63}$$

Furthermore the vanishing of the scalar product between $\rho_{A,\alpha}$ and $\rho_{A,\beta}$ as $u \to 0$ is controlled by $h_{k_0}$ :

$$\frac{\text{Tr}(\rho_{A,\alpha}\rho_{A,\beta})}{\|\rho_{A,\alpha}\|\|\rho_{A,\beta}\|} = \frac{Z_{\alpha|\beta}(\tau)}{\sqrt{Z_{\alpha|\alpha}(\tau)Z_{\beta|\beta}(\tau)}} \underset{u\to 0}{\sim} u^{2h_{k_0}} \left(\frac{s_L(2v)}{8s_L^2(v)}\right)^{2h_{k_0}}. \tag{2.64}$$

## 3  Comparison with numerics and finite-size scaling

### 3.1  Rényi entropy in a quantum Ising chain

In order to compare the theoretical prediction obtained in Section 2 with numerical determinations of the Rényi entropy in a critical lattice model, we have focused on the model that was the most numerically accessible, *i.e.* the Ising spin chain with *free boundary conditions*, with Hamiltonian:

$$H_{\text{free}} = -\sum_{j=1}^{N-1} s_j^x s_{j+1}^x - h \sum_{j=1}^N s_j^z, \tag{3.1}$$

where the $s_j^a$ have their usual definition – they act as Pauli matrices $\sigma^a$ at site $j$ and trivially on the other sites of the system. The chain is taken to have length $L = Na$, where $a$ is the lattice spacing, and $N$ is the number of spins. The *scaling limit* of this system corresponds to taking $N \to \infty$ and $a \to 0$ while keeping the chain length $L$ fixed. Finally, to achieve criticality, the external field $h$ should be set to $h = 1$. We stress that both the *bulk* and the *boundary* of the chain are critical at this point in the parameter space of the model.

A convenient feature of this model is that it can be mapped to a fermionic chain through a Jordan-Wigner (JW) transformation

$$c_k^\dagger = \prod_{j=0}^{k-1} s_j^z s_k^+, \qquad s_k^\pm \equiv \frac{1}{2}(s_k^x \pm i\, s_k^y). \tag{3.2}$$

Once the Hamiltonian of the fermionic chain has been obtained, one proceeds to find a basis of fermionic operators $\eta_i, \eta_i^\dagger$ that diagonalizes it – and still satisfies the standard anti-commutation relations $\{\eta_i, \eta_j^\dagger\} = \delta_{ij}$, *etc.* For free or periodic boundary conditions, the procedure is standard, and we refer the reader to the excellent review [90]. Having found the diagonal fermionic basis $\eta_i, \eta_i^\dagger$ one proceeds to build the correlation matrix $\mathbf{M} \equiv \langle \boldsymbol{\eta} \cdot \boldsymbol{\eta}^\dagger \rangle$ with $\boldsymbol{\eta} \equiv (\eta_1, \ldots, \eta_N, \eta_1^\dagger, \ldots, \eta_N^\dagger)^T$. The eigenvalues of $\mathbf{M}$ are simply related to the values of the entanglement spectrum, and thus one can calculate Rényi entropies for large sizes with an advantageous computational cost that scales as $\mathcal{O}(N)$ with the number $N$ of spins in the system. This method, known in the literature as *Peschel's trick* [91,92], has been employed in several works [93,67] for both free and periodic boundary conditions, and we refer to them for detailed explanations of the implementation.

Due to the JW "strings" of $s^z$ operators in (3.2), the relation between the fermionic and spin reduced density matrices of a given subsystem may be non-trivial [93,94]. For free and periodic BC though, the ground-state wavefunction has a well-defined *parity* of the fermion number, and, as a consequence, the fermionic and spin reduced density matrices of a single interval can be shown to coincide. The Peschel trick fails, however, for the case of fixed BC, where the above feature of the wavefunction no longer holds, as pointed out in [66]. There has been progress, however, in adapting the trick to fixed BC, for the case of an interval touching the boundary [66–68]. Extending the technique to efficiently find the entanglement spectrum for an interval A that does not touch the boundary is still an open problem.

To give concrete expressions to compare with the numerical data, we will quickly review some basic aspects of the CFT description of the critical Ising chain. It is well known that in the critical regime, the scaling limit of the infinite and periodic Ising chains is the Ising CFT, namely the CFT with central charge $c = 1/2$ and an operator spectrum consisting of three primary operators – the identity $\mathbb{I}$, energy $\varepsilon$ and spin operators $\sigma$ – and their descendants [80]. The case of open boundaries is also well understood from the CFT perspective. There are three conformal boundary conditions for the Ising BCFT, which, in the framework of radial quantization on the annulus, allow the construction of the following physical boundary states [59,80]:

$$|f\rangle = |\mathbb{I}\rangle\!\rangle - |\epsilon\rangle\!\rangle \quad \text{(free BC)}, \tag{3.3}$$

$$|\pm\rangle = \frac{1}{\sqrt{2}}|\mathbb{I}\rangle\!\rangle + \frac{1}{\sqrt{2}}|\epsilon\rangle\!\rangle \pm \frac{1}{2^{1/4}}|\sigma\rangle\!\rangle \quad \text{(fixed BC)}, \tag{3.4}$$

where $|i\rangle\!\rangle$ denotes the Ishibashi state [59,95] corresponding to the primary operator $i$. The physical boundary states $|\alpha\rangle$ are in one-to-one correspondence with the primary fields of the bulk CFT[4]: $|f\rangle \leftrightarrow \sigma$ and $|\pm\rangle \leftrightarrow \mathbb{I}/\varepsilon$. The annulus partition function for the Ising BCFT is compactly written in terms of Jacobi theta functions for all diagonal choices of BCs $(\alpha|\alpha)$ and, in consequence, in terms of the parameter $x$ defined in Section 2

$$Z_{f|f}(\tau) = \sqrt{\frac{\theta_3(\tau)}{\eta(\tau)}} = 2^{1/6}\left(x\sqrt{1-x^2}\right)^{-\frac{1}{12}}. \tag{3.5}$$

and

$$Z_{+|+}(\tau) = Z_{-|-}(\tau) = \frac{\sqrt{\theta_3(\tau)} + \sqrt{\theta_4(\tau)}}{2\sqrt{\eta(\tau)}} = 2^{1/6}\frac{1+x^{\frac{1}{4}}}{2}\left(x\sqrt{1-x^2}\right)^{-\frac{1}{12}}. \tag{3.6}$$

These relations allow us to express the orbifold two-point correlator on the disk in an elementary way:

$$\langle\sigma(0,0)\sigma(x,\bar{x})\rangle_{\mathbb{D}}^{(f,f)} = \left[|x|^2(1-|x|^2)\right]^{-\frac{1}{8}}, \tag{3.7}$$

$$\langle\sigma(0,0)\sigma(x,\bar{x})\rangle_{\mathbb{D}}^{(+,+)} = \frac{1+|x|^{\frac{1}{4}}}{2}\left[|x|^2(1-|x|^2)\right]^{-\frac{1}{8}}, \tag{3.8}$$

---

[4]This statement is strictly true if the bulk CFT is diagonal, see [96] for a detailed discussion.

which is, of course, very convenient for numerical checks. Note that the $\mathbb{Z}_2$ orbifold of the Ising model is equivalent to a special case of the critical Ashkin-Teller model [97]. Therefore, the CFT we are considering here is nothing but the $\mathbb{Z}_2$ orbifold of a free boson. This might explain why the above two-point functions end up being so simple.

Recall that the lattice operator $\widehat{\sigma}_{m,n}$ labelled by discrete indices is described in the scaling limit by $\widehat{\sigma}_{m,n} \sim A\, a^{h_\sigma + \bar{h}_\sigma}\, \sigma(w, \bar{w})$, where $w = am + ian$, and $A$ is a non-universal amplitude. Hence, to obtain collapsed data for various chain lengths, it will be convenient to introduce

$$\mathcal{G}_2^\alpha([j,k]) \equiv \widehat{S}_2^\alpha([j,k]) - \frac{1}{8}\log\left(\frac{2N}{\pi}\right) = -\log\langle\widehat{\sigma}_{j,0}\widehat{\sigma}_{k,0}\rangle_{\mathbb{S}_L}^{(\alpha,\alpha)} - \frac{1}{8}\log\left(\frac{2N}{\pi}\right) , \qquad (3.9)$$

so that, in the scaling limit, one expects from (2.11)

$$\mathcal{G}_2^\alpha([j,k]) \sim -\log\langle\sigma(u,\bar{u})\sigma(v,\bar{v})\rangle_{\mathbb{S}_L}^{(\alpha,\alpha)} - \frac{1}{8}\log\left(\frac{2L}{\pi}\right) \qquad (3.10)$$

$$\sim -\log\langle\sigma(u,\bar{u})\sigma(v,\bar{v})\rangle_{\mathbb{D}}^{(\alpha,\alpha)} + \frac{1}{8}\log\left[\sin\frac{\pi(u+v)}{2L}\right] , \qquad (3.11)$$

where $u = aj$ and $v = ak$. We remind that the length of the interval is given by $\ell = v - u = am$ with $m = k - j$, and emphasize that the entanglement is considered for the *ground state* of the free BC Ising chain.

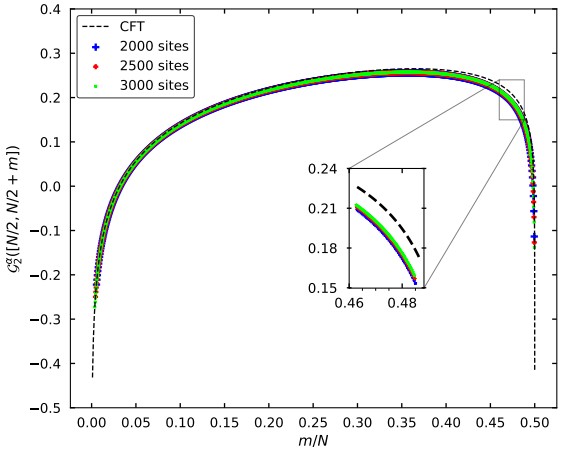

Figure 3: Plot of shifted Rényi entropy $\mathcal{G}_2^f([N/2, N/2 + m])$ for the Ising chain with free BC, against the scaled interval size $m/N$. The deviations from the theoretical predictions are stronger as the interval grows closer to the boundary.

To graphically emphasize the agreement between the fermionic chain data and the theoretical prediction, we have looked at two ways of "growing" the interval length $\ell$. In Figure 3, we have considered an interval that starts in the middle of the chain and grows towards one end. This corresponds on the lattice to applying the first twist operator to the middle of the chain, and the second one progressively closer to the right boundary. Since twist operators are placed *between lattice sites*, one should consider even system sizes. The curves of Figure 4, follow the dependence of the $S_2$ entropy as the interval length $\ell$ is grown equidistantly from the middle of the chain towards the boundaries. We see, in both cases, that the agreement with the CFT prediction is very good, although, as we will detail below, one needs to consider unusually large system sizes to reach it.

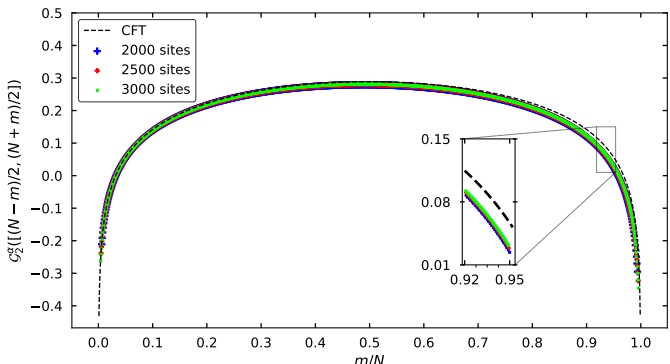

Figure 4: Plot of shifted Rényi entropy $\mathcal{G}_2^f([N-m)/2, (N+m)/2])$ for the Ising chain with free BC, against the scaled interval size $m/N$. The deviations from the theoretical predictions are stronger as the interval is grown towards the boundaries.

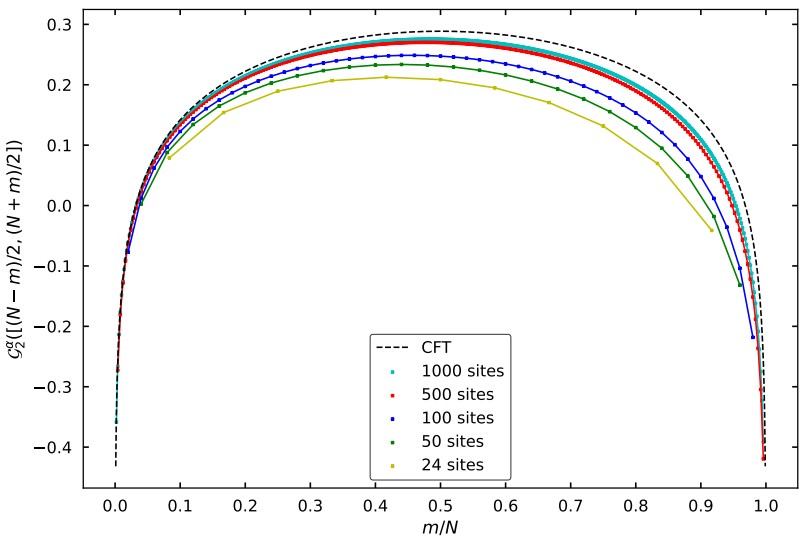

Figure 5: Plot of shifted Rényi entropy $\mathcal{G}_2^f([(N-m)/2, (N+m)/2])$ for a wide range of system sizes. The sizes typically accessible to exact diagonalization ($N \sim 10$) or DMRG methods ($N \sim 10^3$) suffer from large finite-size effects.

## 3.2   Finite-size effects

There is a plethora of sources of finite-size corrections to the orbifold CFT result calculated in Section 2. One should be aware of corrections from irrelevant *bulk* and *boundary* deformations of the Hamiltonian [98], as well as unusual corrections to scaling as analysed in [99,100]. Finally, one should generically worry about parity effects [101] but, in agreement with [102], we have found no such corrections in the numerical results.

The strongest corrections, however, come from the subleading scaling of the lattice twist

operators [52]. We remind that the lattice twist operator $\widehat{\sigma}$ can be expressed, in the continuum limit $a \to 0$ as a *local* combination of scaling operators [103]:

$$\widehat{\sigma}_{m,n} = A\, a^{2h_\sigma} \sigma(w, \bar{w}) + B\, a^{2h_{\sigma_\varepsilon}} \sigma_\varepsilon(w, \bar{w}) + \dots \tag{3.12}$$

where the integers $(m, n)$ give the lattice position of the operator as $w = a(m + in)$, and the dots in (3.12) denote the contribution from descendant operators. The amplitudes $A$ and $B$ of the scaling fields are non-universal, and thus cannot be inferred from CFT methods. However, since the expansion (3.12) does not depend on the global properties of the system, it is independent of the choice of BC. Using the exact results for the correlation matrix $\mathbf{M}$ of the fermionic system associated with an infinite Ising chain [8], and the well-known result for the Rényi entropies of an interval of length $\ell$ in an infinite system [3,9], one can find a fit for the values of $A$ and $B$.

Moving on, the *excited twist operator* $\sigma_\varepsilon$ can be defined through point-splitting as [52], [104]:

$$\sigma_\varepsilon(w, \bar{w}) := \lim_{\eta \to w} \left[ (2|\eta - w|)^{2h_\varepsilon} \sigma(w, \bar{w})(\varepsilon(\eta, \bar{\eta}) \otimes \mathbb{I}) \right] . \tag{3.13}$$

This operator has conformal dimensions $h_{\sigma_\varepsilon} = \bar{h}_{\sigma_\varepsilon} = h_\sigma + h_\varepsilon/2$. The expansion (3.12) implies that in our case, the correlator of twist operators on the Ising spin chain with free boundary conditions can be expressed in terms of CFT correlators as:

$$\langle \widehat{\sigma}_{j,0}\, \widehat{\sigma}_{k,0} \rangle_N^{(f,f)} = A^2\, a^{4h_\sigma} \langle \sigma(u, \bar{u})\sigma(v, \bar{v}) \rangle_{\mathbb{S}_L}^{(f,f)}$$
$$+ AB\, a^{4h_\sigma + h_\varepsilon} \left[ \langle \sigma_\varepsilon(u, \bar{u})\sigma(v, \bar{v}) \rangle_{\mathbb{S}_L}^{(f,f)} + \langle \sigma(u, \bar{u})\sigma_\varepsilon(v, \bar{v}) \rangle_{\mathbb{S}_L}^{(f,f)} \right]$$
$$+ \dots \tag{3.14}$$

Using the map (2.3), and recalling that $L = Na$, we get

$$\langle \widehat{\sigma}_{j,0}\, \widehat{\sigma}_{k,0} \rangle_N^{(f,f)} = A^2 \left( \frac{\pi}{2N} \right)^{c/4} \frac{\langle \sigma(0,0)\sigma(x, \bar{x}) \rangle_{\mathbb{D}}^{(f,f)}}{\left[ \sin \frac{\pi(u+v)}{2L} \right]^{c/4}} + AB \left( \frac{\pi}{2N} \right)^{c/4 + h_\varepsilon} G_L(u, v) + \dots \tag{3.15}$$

where $u = aj$ and $v = ak$ are the physical positions of the twist operators, and $x$ is given by (2.4). The first term in the right-hand side of (3.15) corresponds to the two-point function (2.29), whereas the function $G_L(u, v)$ in the second term is defined as

$$G_L(u, v) = \frac{y^{-h_\varepsilon} \langle \sigma_\varepsilon(0,0)\sigma(x, \bar{x}) \rangle_{\mathbb{D}}^{(f,f)} + y^{h_\varepsilon} \langle \sigma(0,0)\sigma_\varepsilon(x, \bar{x}) \rangle_{\mathbb{D}}^{(f,f)}}{\left[ \sin \frac{\pi(u+v)}{2L} \right]^{c/4 + h_\varepsilon}} , \tag{3.16}$$

with $y = \sin \frac{\pi u}{L} / \sin \frac{\pi(u+v)}{2L}$. The exact determination of the function $G_L(u, v)$ is beyond the scope of the present work – for instance, through a conformal mapping, it would imply the calculation of the one-point function of the energy operator on the annulus. Since, in the Ising CFT, we have $h_\varepsilon = 1/2$, this second term gives a correction of order $1/\sqrt{N}$ to the Rényi entropy predicted by (1.8), which is in agreement with the results of [71,99,66]. This is a very significant correction, and it shows why the system sizes accessible through exact diagonalization (limited to $N < 30$) are not sufficient to separate the leading contribution from its subleading corrections.

To show the dramatic effect of this term, Figure 5 contains a comparison of the collapse for diverse system sizes. As noticed in other works [63], where DMRG methods were used, system sizes of $N \sim 100$ are not enough to satisfyingly collapse the data.

Furthermore, the module organization of the fields in the orbifold CFT implies that the scaling exponents of finite-size corrections are half-integer spaced: there will be contributions

both at relative order $\mathcal{O}(N^{-1})$, and $\mathcal{O}(N^{-3/2})$, and so on. This increases the difficulty of a finite-size analysis, since there are more terms with significant contributions for the system sizes that are numerically accessible. To illustrate this, we give in Figure 6 a plot of the subleading contributions to the lattice twist correlator

$$F_{\text{subleading}}(j,k) = \langle \widehat{\sigma}_{j,0} \, \widehat{\sigma}_{k,0} \rangle_N^{(f,f)} - A^2 \left( \frac{\pi}{2N} \right)^{c/4} \frac{\langle \sigma(0,0)\sigma(x,\bar{x}) \rangle_{\mathbb{D}}^{(f,f)}}{\left[ \sin \frac{\pi(u+v)}{2L} \right]^{c/4}} \, . \qquad (3.17)$$

The plot shows that even at $N \sim 10^3$ the collapse is not perfect.

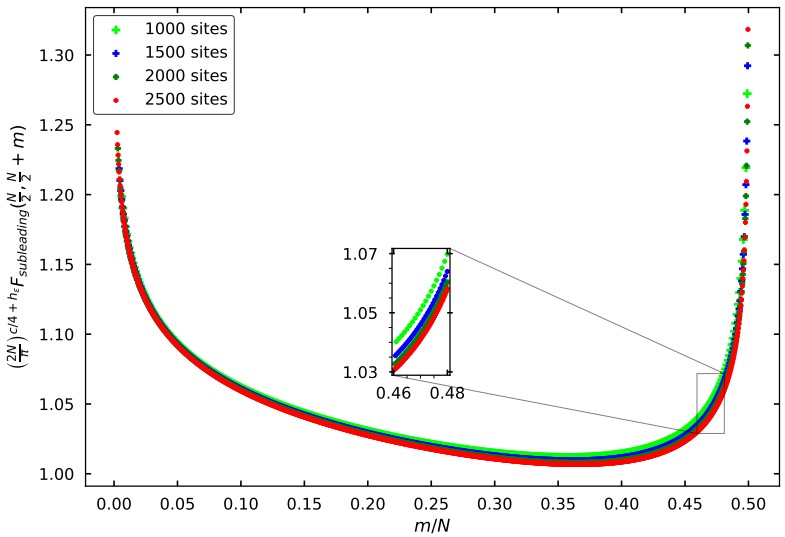

Figure 6: The rescaled subleading contribution $(2N/\pi)^{c/4+h_\varepsilon} F_{\text{subleading}}$ to the lattice two-point function of twist fields, for an interval starting in the middle of the chain and growing towards one boundary. The plot shows that even at large system sizes, the finite-size corrections are significant.

# 4   Conclusion

In this article, we have reported exact results for the Rényi entropy $S_2$ of a single interval in the ground state of a 1D critical system with open boundaries, assuming the same boundary conditions on both sides. This amounts to computing the two-point function of twist operators in the unit disk with diagonal BCs $(\alpha, \alpha)$ in the $\mathbb{Z}_2$ orbifold framework.

By constructing a biholomorphic mapping from the annulus to the two-sheeted disk, we have managed to express the orbifold two-point correlator of twist fields in terms of the annulus partition function of the mother CFT. We have also detailed in the Appendix an alternative derivation of the result for minimal CFTs in the $A$-series.

We have numerically checked the CFT result, and found good agreement with Ising spin chain data, for free BC. It was, however, necessary to achieve large system sizes for this purpose, as the finite-size corrections decayed slowly ($\sim N^{-h_\varepsilon}$ relatively to the dominant term) with the number of sites, as opposed to the case of an interval in a periodic chain, where this decay is of order $N^{-2h_\varepsilon}$ (see [52] for instance). Checking the result for other models and BCs could be achievable through more sophisticated numerical techniques, like (adaptations of) the DMRG approach (see [63, 105]).

A natural extension would be to consider the second Rényi entropy for a system with different conformal BCs on each side of the strip. However, this situation adds the extra complication of insertions of boundary condition changing operators (BCCOs) into the correlator of twist operators [85], thus requiring a calculation of the four-point function of boundary operators on the two-sheeted disk.

# Acknowledgments

The authors are thankful to J.M. Stéphan for suggesting the Peschel trick to study the entanglement entropy of the Ising chain. We also acknowledge P. Calabrese, J. Dubail, R. Santachiara, E. Tonni, M. Rajabpour and J. Viti for their useful comments on the manuscript.

# A   Conventions and identities for elliptic functions

In this Appendix, we fix our notations and conventions for elliptic functions.

## A.1   Jacobi theta functions

We use the following conventions for the Jacobi theta functions $\theta_i(t|\tau)$ :

$$
\begin{aligned}
\theta_1(t|\tau) &= -i \sum_{r \in \mathbb{Z}+1/2} (-1)^{r-1/2} y^r q^{r^2/2}, & \theta_2(t|\tau) &= \sum_{r \in \mathbb{Z}+1/2} y^r q^{r^2/2}, \\
\theta_3(t|\tau) &= \sum_{n \in \mathbb{Z}} y^n q^{n^2/2}, & \theta_4(t|\tau) &= \sum_{n \in \mathbb{Z}} (-1)^n y^n q^{n^2/2},
\end{aligned}
\tag{A.1}
$$

where $q = e^{2i\pi\tau}$ and $y = e^{2i\pi t}$. Here, $t$ is a complex variable and $\tau$ a complex parameter living in the upper half-plane. Theta functions have a single zero, located at $z = 0, 1/2, (1+\tau)/2$ and $\tau/2$, respectively. They have no pole. Using Jacobi's triple product identity one can rewrite them as

$$
\theta_1(t|\tau) = -iy^{1/2}q^{1/8} \prod_{n=1}^{\infty}(1-q^n) \prod_{n=0}^{\infty}(1-yq^{n+1})(1-y^{-1}q^n) ,
$$

$$
\theta_2(t|\tau) = y^{1/2}q^{1/8} \prod_{n=1}^{\infty}(1-q^n) \prod_{n=0}^{\infty}(1+yq^{n+1})(1+y^{-1}q^n) ,
$$

$$
\theta_3(t|\tau) = \prod_{n=1}^{\infty}(1-q^n) \prod_{r\in\mathbb{N}+1/2}^{\infty}(1+yq^r)(1+y^{-1}q^r) , \tag{A.2}
$$

$$
\theta_4(t|\tau) = \prod_{n=1}^{\infty}(1-q^n) \prod_{r\in\mathbb{N}+1/2}^{\infty}(1-yq^r)(1-y^{-1}q^r) .
$$

They satisfy the following half-period relations

$$
\begin{aligned}
\theta_1(t|\tau) &= -i\, e^{i\pi(t+\tau/4)}\, \theta_4(t+\tau/2|\tau) , \\
\theta_2(t|\tau) &= e^{i\pi(t+\tau/4)}\, \theta_3(t+\tau/2|\tau) , \\
\theta_3(t|\tau) &= e^{i\pi(t+\tau/4)}\, \theta_2(t+\tau/2|\tau) , \\
\theta_4(t|\tau) &= -i\, e^{i\pi(t+\tau/4)}\, \theta_1(t+\tau/2|\tau) .
\end{aligned} \tag{A.3}
$$

The functions $\theta_i(0|\tau) \equiv \theta_i(\tau)$ are

$$
\theta_2(\tau) = \sum_{n\in\mathbb{Z}} q^{(n+1/2)^2/2} = 2q^{1/8}\prod_{n=1}^{\infty}(1-q^n)(1+q^n)^2 ,
$$

$$
\theta_3(\tau) = \sum_{n\in\mathbb{Z}} q^{n^2/2} \quad = \prod_{n=1}^{\infty}(1-q^n)\left(1+q^{n-1/2}\right)^2 , \tag{A.4}
$$

$$
\theta_4(\tau) = \sum_{n\in\mathbb{Z}}(-1)^n q^{n^2/2} = \prod_{n=1}^{\infty}(1-q^n)\left(1-q^{n-1/2}\right)^2 .
$$

Finally, we note the following relations

$$
\theta_3^4(\tau) = \theta_2^4(\tau) + \theta_4^4(\tau) , \qquad 2\eta^3(\tau) = \theta_2(\tau)\theta_3(\tau)\theta_4(\tau) , \tag{A.5}
$$

where $\eta(\tau)$ is the Dedekind eta function :

$$
\eta(\tau) = q^{\frac{1}{24}} \prod_{n=1}^{\infty}(1-q^n) . \tag{A.6}
$$

## A.2   Elliptic integral of the first kind

The elliptic integral of the first kind $K(x)$ is given by:

$$
K(x) = \int_0^{\frac{\pi}{2}} \frac{\mathrm{d}\theta}{\sqrt{1-x^2\sin^2\theta}} = \frac{\pi}{2} {}_2F_1\left(\frac{1}{2},\frac{1}{2},1;x^2\right) = \frac{\pi}{2}\theta_3^2(\tau) . \tag{A.7}
$$

This means

$$
x = \frac{\theta_2^2(\tau)}{\theta_3^2(\tau)} . \tag{A.8}
$$

The parameter $x$ is called the elliptic modulus. The inverse relation is

$$q = e^{2i\pi\tau} = \exp\left(-2\pi\frac{K(x')}{K(x)}\right), \qquad x' = \sqrt{1-x^2} \tag{A.9}$$

or equivalently

$$\tau = i\frac{K(\sqrt{1-x^2})}{K(x)}. \tag{A.10}$$

In particular one can check that

$$x(1-x^2)\frac{d\tau}{dx} = \frac{2}{i\pi\theta_3^4(\tau)}. \tag{A.11}$$

## A.3   Weierstrass elliptic function

One possible way to derive the differential equation (2.20) is to express the function $g(t)$ defined in (2.5) in terms of the Weierstrass elliptic function $\wp(t)$. The function $\wp : \mathbb{T}_\tau \to \widehat{\mathbb{C}}$ is defined on the complex torus $\mathbb{T}_\tau = \mathbb{C}/(\mathbb{Z} + \tau\mathbb{Z})$ and takes values in the Riemann sphere $\widehat{\mathbb{C}} = \mathbb{C} \cup \{\infty\}$ :

$$\wp(t) = \frac{1}{t^2} + \sum_{(m,n)\in\mathbb{Z}^2\backslash(0,0)} \frac{1}{(t-m-n\tau)^2} - \frac{1}{(m+n\tau)^2}. \tag{A.12}$$

This is a covering map of the two-sphere $\widehat{\mathbb{C}}$ with 4 ramification points :

$$e_1 = \wp(1/2), \qquad e_2 = \wp\left(\frac{1+\tau}{2}\right), \qquad e_3 = \wp\left(\frac{\tau}{2}\right), \qquad \infty = \wp(0). \tag{A.13}$$

The lattice roots $e_i$ can be expressed in terms of the theta functions as :

$$e_1 = \frac{\pi^2}{3}\left(\theta_2^4(\tau) + 2\theta_4^4(\tau)\right), \quad e_2 = \frac{\pi^2}{3}\left(\theta_2^4(\tau) - \theta_4^4(\tau)\right), \quad e_3 = -\frac{\pi^2}{3}\left(2\theta_2^4(\tau) + \theta_4^4(\tau)\right). \tag{A.14}$$

The function $g(t)$ as defined in (2.5) is simply the composition of $\wp(t)$ with a particular Möbius transformation that sends the ramification points to $0, 1/x, x$ and $\infty$ :

$$g(t) = \frac{1}{x}\frac{\wp(t) - e_3}{e_1 - e_3}, \qquad x = \sqrt{\frac{e_2 - e_3}{e_1 - e_3}} = \left(\frac{\theta_2(\tau)}{\theta_3(\tau)}\right)^2, \tag{A.15}$$

as follows from the fact that $(\wp(t) - e_3)/g(t)$ is constant by virtue of being doubly periodic and holomorphic (*i.e.* with no pole). Now from the differential equation obeyed by $\wp(t)$, namely

$$\wp'^2(t) = 4\left(\wp(t) - e_1\right)\left(\wp(t) - e_2\right)\left(\wp(t) - e_3\right), \tag{A.16}$$

we get

$$\left(\frac{dg}{dt}\right)^2 = -4\pi^2\theta_3^4(\tau)g(g-x)(1-xg), \tag{A.17}$$

from which (2.20) follows.

# B  Alternative derivation of the second Rényi entropy for $A_n$ minimal models

In this Appendix we present an alternative computation of the two-twist correlation function (2.28) based on the mirror trick [59] and BCFT bootstrap methods [85]. The conformal blocks are obtained in terms of the modular characters (see also [52,82]). The other key ingredients are the bulk and bulk-boundary structure constants appearing in the conformal block expansion. We note that the correspondence between conformal blocks and characters has also been employed in the recent work of [106] for the evaluation of twist correlators on manifolds without boundaries.

In order to avoid some technicalities, we restrict our attention to Virasoro minimal models in the $A_n$ series, for which the torus partition function is a diagonal modular invariant. On the unit disk, the mirror trick amounts to replacing the disk by its Schottky double [107], namely a sphere, and bulk fields $\phi(z, \bar{z})$ by a pair of chiral fields, one at position $z$ and the other at its mirror image $1/\bar{z}$ :

$$\phi(z, \bar{z}) \to \phi(z)\, \bar{z}^{-2h} \phi(1/\bar{z})\,. \tag{B.1}$$

Thus we can decompose $\langle \sigma(0,0)\sigma(x,\bar{x})\rangle_{\mathbb{D}}^{(\alpha,\alpha)}$ as a linear combination of conformal blocks on the sphere

$$\langle \sigma(0,0)\sigma(x,\bar{x})\rangle_{\mathbb{D}}^{(\alpha,\alpha)} = \sum_j X_j^\alpha f_j(x,\bar{x})\,, \tag{B.2}$$

$$f_j(x,\bar{x}) = \bar{x}^{-2h_\sigma}\quad \begin{array}{c} \sigma(0)\diagdown \phantom{xx}\diagup \sigma(\infty) \\ \phantom{x}\phi_j \otimes \phi_j \\ \sigma(x)\diagup \phantom{xx}\diagdown \sigma(1/\bar{x}) \end{array}\quad. \tag{B.3}$$

Indeed, for the $\mathbb{Z}_2$ orbifold of a minimal model in the $A_n$ series, the fusion $\sigma \times \sigma$ is of the form [52]

$$\sigma \times \sigma = \sum_{\phi_j\ \text{primary}} \phi_j \otimes \phi_j\,, \tag{B.4}$$

where the sum runs over the primary operators of the mother CFT. We shall denote by $h_j$ the conformal dimension of $\phi_j$ (recall that for $A_n$ minimal models, all primary operators are scalar, so $\bar{h}_j = h_j$). The expansion coefficients $X_j^\alpha$ in (B.2) are obtained in terms of OPE structure constants as

$$X_j^\alpha = C_{\sigma\sigma}^{\phi_j \otimes \phi_j}\, A_{\phi_j \otimes \phi_j}^{(\alpha,\alpha)}, \tag{B.5}$$

which in turn can be expressed as [82]

$$A_{\phi_j \otimes \phi_j}^{(\alpha,\alpha)} = \langle (\phi_j \otimes \phi_j)(0)\rangle_{\mathbb{D}}^{(\alpha,\alpha)} = (\langle \phi_j(0)\rangle_{\mathbb{D}}^\alpha)^2 = \left(A_j^\alpha\right)^2\,, \tag{B.6}$$

$$C_{\sigma\sigma}^{\phi_j \otimes \phi_j} = \langle \sigma(\infty)(\phi_j \otimes \phi_j)(1)\sigma(0)\rangle_{\mathbb{C}} = 2^{-4h_j}\langle \phi_j(-1)\phi_j(1)\rangle_{\mathbb{C}} = 2^{-8h_j}\,, \tag{B.7}$$

so that

$$X_j^\alpha = 2^{-8h_j}\left(A_j^\alpha\right)^2\,. \tag{B.8}$$

The OPE coefficient $A_j^\alpha$ is very much related to coefficients $\Psi_j^\alpha$ appearing in the decomposition of the boundary state $|\alpha\rangle$ in terms of the Ishibashi states $|j\rangle\!\rangle$ :

$$A_j^\alpha = \frac{\Psi_j^\alpha}{\Psi_0^\alpha}, \qquad |\alpha\rangle = \sum_j \Psi_j^\alpha |j\rangle\!\rangle\,. \tag{B.9}$$

For minimal models in the $A_n$ series, these coefficients are given in terms of the modular $S$-matrix elements [108]:

$$\Psi_j^\alpha = \frac{S_{j\alpha}}{\sqrt{S_{0j}}}, \qquad A_j^\alpha = \frac{S_{j\alpha}}{S_{0\alpha}}\sqrt{\frac{S_{00}}{S_{j0}}}, \tag{B.10}$$

where the index 0 corresponds to the identity operator.

Let us turn to the expression of the conformal blocks $f_j$ in terms of the characters of the mother CFT. By a simple rescaling we have

$$f_j(x, \bar{x}) = \mathcal{F}_j(\eta), \qquad \eta = |x|^2, \tag{B.11}$$

where $\mathcal{F}_j(\eta)$ is the standard conformal block

$$\mathcal{F}_j(\eta) = \begin{array}{c} \sigma(0) \\ \diagdown \\ \sigma(\eta) \end{array} \phi_j \otimes \phi_j \begin{array}{c} \sigma(\infty) \\ \diagup \\ \sigma(1) \end{array} . \tag{B.12}$$

These conformal blocks are known [82] to be related to the characters $\chi_j(\tau)$ of the mother theory via

$$\mathcal{F}_j(\eta) = 2^{8h_j - c/3} \left[\eta(1-\eta)\right]^{-c/24} \chi_j(\tau), \qquad \eta = [\theta_2(\tau)/\theta_3(\tau)]^4. \tag{B.13}$$

Assembling the above results, we obtain the expression

$$\langle \sigma(0,0)\sigma(x,\bar{x}) \rangle_{\mathbb{D}}^{(\alpha,\alpha)} = 2^{-\frac{c}{3}} \left[|x|^2(1-|x|^2)\right]^{-\frac{c}{24}} \sum_j (A_j^\alpha)^2 \chi_j(\tau). \tag{B.14}$$

The last step is to relate the above linear combination of characters to the annulus partition function:

$$Z_{\alpha|\alpha}(\tau) = \langle \alpha | e^{i\pi\tau(L_0 + \bar{L}_0 - c/12)} | \alpha \rangle = \sum_j (\Psi_j^\alpha)^2 \langle\!\langle j | e^{i\pi\tau(L_0 + \bar{L}_0 - c/12)} | j \rangle\!\rangle = \sum_j (\Psi_j^\alpha)^2 \chi_j(\tau), \tag{B.15}$$

using (B.9), and $g_\alpha = \Psi_0^\alpha$.

# C Annulus partition function for the compact boson

For the following discussion, it is useful to have in mind a lattice model whose scaling limit is given by the free compact boson – we take for example the six-vertex (6V) model on the square lattice. It is well established (see [109] for instance) that the 6V model with homogeneous Boltzmann weights

is critical in the regime

$$|\Delta| < 1, \qquad \Delta = \frac{a^2 + b^2 - c^2}{2ab}, \tag{C.1}$$

and is described in the scaling limit by a free compact boson with action

$$S[\phi] = \frac{1}{8\pi} \int d^2r \, \partial_\mu \phi \, \partial^\mu \phi \,, \qquad \phi \equiv \phi + 2\pi R \,, \tag{C.2}$$

where the compactification radius is given by $R = \sqrt{(2/\pi)\cos^{-1}\Delta}$. We consider the 6V model on a rectangle of $M \times N$ sites, with periodic boundary conditions in the horizontal direction, and reflecting boundary conditions at the top and bottom edges, for even $M, N$. Any 6V configuration defines (up to a global shift) a height function on the dual lattice, with steps $\pm \pi R$ between neighbouring heights. Since the local arrow flux into each of the boundaries is zero, the height function is constant along each boundary, and it is periodic in the horizontal direction. However, there can be a flux of $2m$ arrows (with $m \in \mathbb{Z}$) going between the two boundaries, and hence the height difference between the boundaries is of the form $2\pi mR$. In the scaling limit $N, M \to \infty$ with $N/M = \mathrm{Im}\,\tau/2$, the height function renormalizes to the free boson $\phi$, and we get

$$Z_{6V}(M/N) \to \sum_{m \in \mathbb{Z}} Z_{\alpha|\alpha+m}(\tau) \,, \tag{C.3}$$

where $\alpha$ is an arbitrary integer, and $Z_{\alpha|\beta}(\tau)$ denotes the partition function of (C.2) on the annulus of Figure 2 with Dirichlet boundary conditions $\phi(x,0) = 2\pi R\alpha$ and $\phi(x, \mathrm{Im}\,\tau/2) = 2\pi R\beta$. A path integral computation gives

$$Z_{\alpha|\beta}(\tau) = \frac{e^{-i\pi R^2(\alpha-\beta)^2/\tau}}{\eta(-1/\tau)} \,. \tag{C.4}$$

Hence, the scaling limit of the 6V partition function is

$$Z_{6V}(M/N) \to Z(\tau) = \frac{\sum_{m \in \mathbb{Z}} e^{-i\pi R^2 m^2/\tau}}{\eta(-1/\tau)} = \frac{\theta_3(-R^2/\tau)}{\eta(-1/\tau)} \,. \tag{C.5}$$

In the geometry of the infinite strip of width $N$ sites, the 6V transfer matrix generates the XXZ spin-chain Hamiltonian

$$H_{\mathrm{XXZ}} = -\sum_{j=1}^{N-1} \left( s_j^x s_{j+1}^x + s_j^x s_{j+1}^x + \Delta s_j^z s_{j+1}^z \right) \,, \tag{C.6}$$

where $s_j^{x,y,z}$ are Pauli matrices acting on site $j$. Reflecting boundary conditions for the 6V model (and thus Dirichlet boundary conditions for the boson) correspond to free boundary conditions on the spins.

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
