# Peer review of "Second Rényi entropy and annulus partition function for one-dimensional quantum critical systems with boundaries"

_SciPost Physics_

## Round 1 · Referee Report · Anonymous (Referee 1) · 2022-1-15

Strengths

1- The paper finds the solution for a well defined old open problem.
2- The organization of the paper is good.
3- The derived equation is supported with numerical calculations

Weaknesses

1- The solution is found just for n=2 Renyi entropy.
2- Numerical calculations are done for just one type of BC.

Report

This paper find a clean and compact formula for the n=2 Renyi entanglement entropy of an interval in an open CFT. This problem has been an open problem for more than 15 years. The paper exploits the methods of the references [49] and [77] to find the main result, i.e. equation 1.8.
Then the paper support the formula by re-driving the previous results that were valid for especial cases. They also check the validity of the formula using numerical calculations on the transverse field Ising chain with open BC.
Although the main problem of calculating Renyi entropy for general n and the von Neumann entropy still remains an open problem, due to the level of the difficulty of the project, I would not consider it as a major weakness.
The paper is written clearly and follows a good logic. I think the paper has enough merit to be published in Scipost after a few minor additions and changes.

Requested changes

1: I guess a factor 2 is missing in equation 1.4 plus some non-universal constant.
2: Fig 2 is not cited in the main text. In addition the caption does not seem self explanatory. In general I think the mapping can be probably explained better by improving this Figure.
3: It would be nice to see limiting cases of equation 2.52.

  • validity: high
  • significance: high
  • originality: high
  • clarity: high
  • formatting: excellent
  • grammar: excellent

Author:  Andrei Rotaru  on 2022-02-21  [id 2229]

(in reply to Report 1 on 2022-01-15)

We thank the referee for their thorough reading of the manuscript, and their useful comments and suggestions. We have adressed them as follows:

1: I guess a factor 2 is missing in equation 1.4 plus some non-universal constant.

  • The factor of 2 was corrected, and we have addressed the origin of the non-universal constant

2: Fig 2 is not cited in the main text. In addition the caption does not seem self explanatory. In general I think the mapping can be probably explained better by improving this Figure.

  • The figure is now cited in the main text, and we have modified it to show the preimages of the points 0 and x under the map g(t)

3: It would be nice to see limiting cases of equation 2.52.

  • We have added two subsections, discussing two interesting limits of the Schatten distance

Anonymous on 2022-02-23  [id 2243]

(in reply to Andrei Rotaru on 2022-02-21 [id 2229])

The authors successfully took care of my comments. I recommend the paper for publication in its current form.

---

## Round 1 · Referee Report · Anonymous (Referee 2) · 2022-1-17

Report

The paper deals with computation the entanglement entropy in a bipartite system where the subregion of interest is a single interval away from the boundary. The authors report an exact computation of the second Re’nyi entropy in the ground state of a 1+1 dimensional critical system, which is based on the evaluation of a two-point function of twist operators on the infinite strip. Choosing suitable boundary conditions, the problem is reduced to the derivation of the two-twist correlation function on the unit disk D without insertions of boundary operators. The final result is expressed in terms of the annulus partition function of the starting CFT. Some numerical checks for the special case of the Ising spin chain are also presented. Fermionic chains of system size up to103 sites are analysed and the finite-size effects are discussed.

The main result of the paper is new and deserves a publication. The exposition is clear, all statements are carefully formulated and supported by explicit computations. The appendix B contains an alternative derivation of the main result. I recommend the publication of the paper in its present form.

  • validity: -
  • significance: -
  • originality: -
  • clarity: -
  • formatting: -
  • grammar: -

Author:  Andrei Rotaru  on 2022-02-21  [id 2228]

(in reply to Report 2 on 2022-01-17)

We thank the referee for their careful reading of the manuscript and their appreciations.

---

## Editorial Decision

resubmitted